# Calcium Signaling in the Cerebellar Radial Glia and Its Association with Morphological Changes during Zebrafish Development

**DOI:** 10.3390/ijms222413509

**Published:** 2021-12-16

**Authors:** Elizabeth Pereida-Jaramillo, Gabriela B. Gómez-González, Angeles Edith Espino-Saldaña, Ataúlfo Martínez-Torres

**Affiliations:** Laboratorio de Neurobiología Molecular y Celular, Departamento de Neurobiología Celular y Molecular, Instituto de Neurobiología, Universidad Nacional Autónoma de México. Campus UNAM Juriquilla, Queretaro CP76230, Mexico; elizabeth.pereida@comunidad.unam.mx (E.P.-J.); gabebyre@gmail.com (G.B.G.-G.); espinoedi@unam.mx (A.E.E.-S.)

**Keywords:** calcium waves, cerebellum, GECI, GFAP, Tol2

## Abstract

Radial glial cells are a distinct non-neuronal cell type that, during development, span the entire width of the brain walls of the ventricular system. They play a central role in the origin and placement of neurons, since their processes form structural scaffolds that guide and facilitate neuronal migration. Furthermore, glutamatergic signaling in the radial glia of the adult cerebellum (i.e., Bergmann glia), is crucial for precise motor coordination. Radial glial cells exhibit spontaneous calcium activity and functional coupling spread calcium waves. However, the origin of calcium activity in relation to the ontogeny of cerebellar radial glia has not been widely explored, and many questions remain unanswered regarding the role of radial glia in brain development in health and disease. In this study we used a combination of whole mount immunofluorescence and calcium imaging in transgenic (gfap-GCaMP6s) zebrafish to determine how development of calcium activity is related to morphological changes of the cerebellum. We found that the morphological changes in cerebellar radial glia are quite dynamic; the cells are remarkably larger and more elaborate in their soma size, process length and numbers after 7 days post fertilization. Spontaneous calcium events were scarce during the first 3 days of development and calcium waves appeared on day 5, which is associated with the onset of more complex morphologies of radial glia. Blockage of gap junction coupling inhibited the propagation of calcium waves, but not basal local calcium activity. This work establishes crucial clues in radial glia organization, morphology and calcium signaling during development and provides insight into its role in complex behavioral paradigms.

## 1. Introduction

The cerebellum is involved in sensorimotor processing function, and recent evidence also suggests its role in cognition and emotion [1,2]. Damage to the cerebellum may result in a series of motor disturbances, such as ataxia, and it has been argued that cerebellar dysfunction contributes to non-motor conditions, including autism [3,4,5]. Considering all the clinical implications of cerebellar function and that its defects can induce motor and other neurological conditions by interacting with diverse brain areas, there is emerging interest in studying the development and functional organization of the cerebellum.

The overall organization and cell types of the cerebellum are highly conserved from fish to mammals [6,7,8]. Radial glia cells are evolutionarily conserved embryonic neural stem cells that, in the adult cerebellum, remain as Bergmann glia [9,10]. The somas of the radial glia are aligned along the Purkinje cell layer, and, during early development, they extend thin processes toward the pia through which new neurons migrate to the cortical surface [11]. Radial glia cells are also transformed into multiciliate ependymal cells, and in adult fish they remain as neural stem cells [12,13].

Despite the fundamental role of radial glia for cerebellar development and function, little attention has been paid to their morphological and functional changes during early fish development. In contrast, more information has been gathered about the neuronal component. For example, neuronal differentiation in the fish cerebellum begins at 3 days post fertilization (dpf) and layered stratification occurs at 5 dpf [14]. At 4–7 dpf, Purkinje cells process sensory stimuli, such as light and touch, and at 5 dpf, larvae start to hunt and capture prey [15]. In addition, between 6–8 dpf, complex behaviors linked to learning movement are already encoded in the cerebellum [16]. Zebrafish is an attractive experimental model for understanding morphological and functional changes of cells during early brain development, and many studies in this model have focused on the organization and function of neurons and neuronal circuits [8,17]. However, since the discovery that the adult zebrafish brain repairs lesions after injury by inducing radial glia proliferation, new interest in the functional characteristics of these cells has emerged [18,19,20].

Similar to astrocytes, radial glia cells present calcium events called calcium waves that spread across several cells [21,22,23]. Calcium waves occur spontaneously, but require the expression of several key molecular components: the gap junction protein connexin 43, P2Y1 ATP receptors and intracellular inositol-3-phosphate mediated calcium release [24,25]. In the mouse embryo brain cortex, calcium waves from radial glia increase their frequency during neurogenesis, and, when calcium waves are blocked, ventricular zone cell proliferation is reduced [23]. In the adult mammal cerebellum, Bergmann glia shows concerted calcium activity during locomotor performance [26]. In addition, soluble substances, such as growth factors, positively regulate calcium waves, whereas inflammatory cytokines disrupt them [27].

Astrocytes and radial glia sense and modulate neuronal activity, indicating that neurons and glia reciprocally communicate [28,29]. While the origin and functional characteristics of calcium signaling are constantly investigated, the functional consequences for neuronal and glial circuits remain poorly understood. In this study, we constructed a transgenic zebrafish that expresses the genetically encoded calcium sensor GCaMP6s in radial glia to study the ontogenic appearance of calcium signaling during larval cerebellar development. The results showed that cerebellar radial glia display spontaneous calcium elevations even before neuronal circuit formation, and higher calcium activity coincided with an increase in radial glia morphological complexity. Finally, we show that pharmacological blockage of gap junctions inhibits the propagation of calcium waves.

## 2. Results

### 2.1. Distribution of Radial Glia in the Larval Zebrafish Cerebellum

To monitor the morphological changes of cerebellar radial glia during zebrafish development, we focused on larvae at 3, 5 and 7 dpf. GFAP-labeled radial glia cells were surveyed in optical sections obtained by confocal microscopy (N = 5 larvae per age). For reference, we divided the cerebellum into three sections: dorsal, medial, and ventral (Figure 1). At 3 dpf, most of the radial glia cells were vertically aligned along the antero-posterior axis and homogeneously distributed along the cerebellum (Figure 1B). The radial glia somas faced the inner-posterior side of the cerebellum, while their end-feet were pointed toward the anterior part, where they formed a continuum reaching the anterior edge of the cerebellar lobe. The ventricle is evident in the most dorsal section of the cerebellum. In the hindbrain, the radial glia somas faced the dorsal side, while the processes projected toward the ventral section (Figure 1B). In contrast, at 5 and 7 dpf (Figure 1), when the corpus cerebelli (CCe) and the auricle were already present, the radial glia cells adopted diverse morphologies. Radial glia distribution was similar between 5 and 7 dpf. At these times, the valvula cerebelli (Va) presented radial glia with long and dense processes aligned along the midline (Figure 1B). At 7 dpf, in the dorsal and medial section, the radial glia somas were located along the midline and projected their process to the lateral section, thus defining the caudal region of the Va (Figure 1B, yellow asterisks).

In the bilateral CCe, radial glia cells were aligned along the medial-lateral axis (Figure 1B, 7 dpf, upper panel, white dashed line). The somas were in the dorsal CCe and slightly bent inwardly while projecting toward the ventral section. In the medial section of the eminentia granularis (EG), the radial glia somas were tightly organized, and their processes bent inwardly (Figure 1B, blue dashed line), while, in the ventral EG, the radial glia somas were situated in the medial section and the processes were straight. At 7 dpf, in the caudal lobule (LCa), the radial glia cell morphology was homogeneous, with somas localized dorsally and straight processes projected ventrally (Figure 1B, gray dashed line).

### 2.2. Morphometric Analysis of Radial Glia during Development

Individual radial glia cells were analyzed to determine their morphology at 3, 5 and 7 dpf based on the GFAP labeling (*n* = 64–70 cells per group, N = 5 larvae, Figure 2A). To classify the different morphologies, we identified the number of processes per cell, the length of each process and the soma area. The cerebellum was divided into two regions at 3 dpf (lateral and central). At 5 and 7 dpf, we assessed the distribution of the morphologies in relation to the four regions of the cerebellum: Va, CCe, EG and LCa. The descriptive statistics obtained at different ages are summarized in Table 1, and the descriptive statistics per cerebellar region at each analyzed age are shown in Table 2. The comparison between the morphologies among the different cerebellar regions at each analyzed age did not show significant differences in most cases (total statistical results are shown in Appendix A), indicating little variability in radial glia morphology at these ages (Figure 2B). However, we found some exceptions: at 3 dpf, the soma area was larger in the radial glia from the lateral zone than from the central zone (21 ± 1.5 µm^2^ vs. 26 ± 2.0 µm^2^, respectively), and, at 5 dpf, the process length was larger in the LCa than in the Va (67.7 ± 4.7 µm vs. 52.3 ± 3.3 µm).

Due to the fast growth rate during the early stages of development, we found a remarkable variability in radial glia morphologies in all the parameters at 3, 5 and 7 dpf (adjusted *p*-value, α 0.05, Kruskal–Wallis test, Dunn’s post hoc test, Table 1, Figure 2C). A rapid increase in the cell soma area and process length from 3 (soma 17.89 ± 0.97 µm^2^; process length 23.28 ± 1.19 µm) to 5 dpf (soma 59.53 ± 2.17 µm^2^; process length 34.87 ± 2.27 µm) (*p*-value, <0.001 ***, in both) was followed by a significant reduction in the process length at 7 dpf (31.13 ± 1.90 µm) compared to 5 dpf (34.87 ± 2.27 µm) (*p*-value, <0.001 ***). We also observed that the cell soma area was reduced from 5 dpf (59.53 ± 2.17 µm^2^) to 7 dpf (34.37 ± 1.56 µm^2^) (*p*-value, >0.99, ns). Interestingly, the number of processes per radial glia decreased from an average of 2.16 ± 0.12 µm (at 3 dpf) to 1.48 ± 0.09 µm at 5 dpf (*p*-value, 0.007 **), and twice fold to 3.99 ± 0.31 µm at 7 dpf (*p*-value, <0.001 ***). 

### 2.3. Calcium Signaling in Radial Glia during Development

After we established that the radial glia increased their morphological complexity during the first days of development, we assessed if this complexity was accompanied by differences in calcium activity. In order to record the activity, we used high-speed fluorescence microscopy in agar-embedded larvae with radial glia expression of GCaMP6s (Appendix A) and at 5 and 7 dpf we divided the cerebellum into two regions: the dorsal region (field 1), where the somas and processes of the EG, CCe and LCa are located, and the ventral region (field 2), where somas and few processes of the Va and EG are located. At 3 dpf only the CCe and Va were evident, and the activity was recorded in fields 1 and 2 (Appendix A).

Figure 3A shows sample fluorescence images of basal calcium activity during a period of 4.16 min in field 1 and field 2 at 5 dpf. Most of the changes in activity were detected in the cell somas (white dashed line), although some processes showed activity as well (red arrowheads). Thus, our set-up effectively detected basal calcium activity in the whole cerebellum of the paralyzed fish. Regions of interest were detected with the astrocyte quantification and analysis (AQuA) software, and the radial glia soma and processes were covered with a random color mask (Appendix A).

Changes in fluorescence over time (ΔF/F0) were obtained from time-lapse movies (Figure 3A,B) and multiple comparations between groups were performed. First, we found that, at 3 dpf, the number of calcium events in fields 1 and 2 was significantly different (7.4 ± 0.36/4 min vs. 5.50 ± 0.23/4 min, *p*-value < 0.001 ***), at 5 dpf no difference was found between the number of events in fields 1 and 2, (6.15 ± 0.21/4 min vs. 6.79 ± 0.13/4 min), and, at 7 dpf, the number of calcium events was again significantly different between fields 1 and 2 (8.4 ± 0.26/4 min vs. 7.29 ± 0.21/4 min, *p*-value < 0.05 *). Then we compared the calcium events between stages. We noticed an increase in the number of events from 5 to 7 dpf (field 1: 3 dpf, 7.4 ± 0.36/4 min vs. 5 dpf, 6.15 ± 0.21/4 min *p*-value < 0.05 * vs. 7 dpf, 8.43 ± 0.26/4 min *p*-value > 0.05 ns, and field 2: 3 dpf, 5.50 ± 0.23/4 min vs. 5 dpf, 6.79 ± 0.13/4 min vs. 7 dpf, 7.29 ± 0.21 *p*-value = 0.001 ***, Figure 3C). 

Next, we determined amplitude variations over time, duration and rise time of the calcium events. We found that, in field 1, the calcium event amplitude was larger at 5 dpf (1.65 ± 0.04 ΔF/F0) and 7 dpf (1.48 ± 0.03 ΔF/F0) than at 3 dpf (1.34 ± 0.06 ΔF/F0) *p*-value < 0.05 * and <0.001 ***, while, in field 2, the amplitude remained constant (3 dpf 1.77 ± 0.04 ΔF/F0, 5 dpf 1.83 ± 0.02 ΔF/F0, 7 dpf 1.73 ± 0.02 ΔF/F0, Figure 3D). The duration of the calcium events was longer in field 2 than in field 1 at 3 dpf (field 1: 6.94 ± 0.47 s vs. field 2: 10.46 ± 0.46 s, *p*-value < 0.001 ***) however, the durations were similar between fields at 5 dpf (field 1: 11.38 ± 0.43 s vs. field 2: 11.25 ± 0.24 s) and at 7 dpf (field 1: 9.27 ± 0.35 s vs. field 2: 9.96 ± 0.24 s) *p*-value > 0.05 ns. The duration of calcium events was different between analyzed ages in field 1. A sharp increase was observed from 3 to 5 dpf, followed by a reduction at 7 dpf (field 1: 3 dpf, 6.94 ± 0.47 s vs. 5 dpf, 11.38 ± 0.43 s and 7 dpf, 9.27 ± 0.35 s *p*-value < 0.001 ***; Figure 3E). We next explored the rise time of the calcium events. We did not find differences between rise time in field 1 and field 2 at 3 dpf (2.64 ± 0.19 s vs. 3.05 ± 0.15 s). The rise time was faster in field 1 at 3 dpf (2.64 ± 0.19 s) compared to 5 dpf (4.11 ± 0.18 s) *p*-value < 0.001 *** and 7 dpf (3.40 ± 0.15 s) *p*-value < 0.05 *. In contrast, we did not find differences in rise time in field 2 (3 dpf, 3.05 ± 0.15 s; 5 dpf, 3.38 ± 0.08 s; 7 dpf, 3.38 ± 0.11 s; Figure 3F).

Finally, to understand how the dynamics of calcium oscillations changed during development, we determined the interactions between the number of simultaneous calcium transient events over time in field 1 and field 2 at 3, 5 and 7 dpf. At 3 dpf, few synchronized events occurred during the recording time (an average of 3 and 6 events in field 1 and 2, respectively) which occurred between 1–2 pairs of radial glia cells in field 1, and about 3–4 radial glia cells in field 2. At 5 dpf, the synchronization events were more frequent in field 1, which showed an average of 10 events with 2–10 synchronized radial glia cells, while field 2 showed a peak of 18 events with 20 synchronized radial glia cells. At 7 dpf, synchronized radial glia cells, were slightly higher than at 5 dpf. In field 1 the peak of synchronized activity consisted of 20 events in which 16 radial glia cells were synchronized, while, in field 2, the peak of synchronized activity was of 10 events in which 20 radial glia cells were synchronized (Figure 3G). These data indicate that the calcium events and their synchronization to form functional networks correlate with the development of morphological complexity in radial glia cells.

### 2.4. Calcium Waves in the Developing Cerebellum

Calcium elevations appear spontaneously affecting neural circuits and vasculature. In mammals, calcium waves propagate through radial glia in the embryonic ventricular zone in response to electrical or mechanical stimuli [23]. After we showed that many calcium events occur spontaneously in the developing cerebellum in zebrafish, we evaluated their synchronization to generate calcium waves.

Using time-lapse calcium imaging, we observed spontaneous calcium activity in the radial glia in field 1 occurring in small cell clusters at 3 dpf. At this age, 46 events per minute were observed in 149.5 ± 2.25 cells and the activity spread only to small cell clusters (no more than 4.94 µm). Calcium waves were effectively detected from 5 dpf in the cerebellum. At this age, 735.79 events per minute were observed and 2391.30 ± 39.24 cells on average were synchronized to form a calcium wave. Unexpectedly, the calcium activity was reduced at 7 dpf to 453.74 events per minute and 1474.70 ± 7.6 cells synchronized to form a calcium wave. Figure 4 shows raster plots of the events detected per age; the plots show evidence of the synchronized events of multiple cells during the recording time.

The spread of calcium waves was directional. They first started in the most posterior end of the cerebellum and spread to the anterior region (Figure 4B). Indeed, calcium waves seem to be global events that start in the hindbrain, pass through the cerebellum and continue to the midbrain (Appendix A); however, in this report, we focused only on the events recorded in the cerebellum. In this region, the calcium activity was first detected in the somas of the radial glia, then it spread to the cell processes reaching the end-feet at the pial surface. Eventually the activity disappeared, first form the processes and finally from the soma (Figure 4C) before a new wave arrived. To further classify the characteristics of the calcium activity during cerebellar development, we divided the events in three groups: (a) non-spreading events (<20 µm), (b) short-distance spreading events (20–60 µm), and (c) long-distance spreading events (> 60 µm). At 3 dpf, all the events detected were non-spreading events (897 events 4.29 ± 0.48 µm distance). However, at 5 dpf 127, events spread were short distance 32.21 ± 10.46 µm and, 3437 events spread long distance (138.71 ± 18.75 µm); these calcium waves propagated with low frequency, but recruited most of the cells of the cerebellum. Finally, at 7 dpf 8820 events were non-spreading events (0.21 ± 0.01 µm), 25 events spread were short distance (30.34 ± 1.77 µm) and 2500 events spread long distance (101.65 ± 19.20 µm) (Figure 4D). The velocity of the events fluctuated over time, the maximal velocity at 3 dpf was 10 µm/s and increased at 5 and 7 dpf to around 40–60 µm/s (Figure 4E). Collectively, our data show patterns of calcium activity associated with the developmental morphological complexity of radial glia in the cerebellum.

### 2.5. Effect of Gap Junction Blockage on Calcium Waves

Many different models have been proposed to understand the molecular components that give rise to calcium waves. Cell coupling via gap junctions is central to explaining the formation of calcium waves, since gap junctions are permeable to small molecules, including calcium and inositol-3-phospate [23,30,31,32]. Thus, we analyzed the spread of calcium waves after exposing the fish to heptanol, a well-known gap junction blocker [33]. As in the previous section, the calcium events were analyzed with AQuA software (N = 6 larvae per condition Figure 5A).

Basal cerebellar calcium activity of a fish treated with 500 µM heptanol for 30 min is shown in Appendix A, and Figure 5 shows sample images selected from the same video to illustrate maximal calcium wave spread in controls (Figure 5A panel 1) and short-distance propagation in heptanol treated fish (Figure 5A panel 2). This series of images shows that heptanol severely affects the dimensions of the calcium wave activity as compared to a non-treated fish, strongly suggesting the role of gap junctions in the propagation of calcium waves. Further analysis of the dynamics of calcium waves showed differences in synchrony of calcium events (Figure 5B, control 21.96 ± 0.10/19.5 min vs. heptanol 3.62 ± 0.06/19.5 min, *p*-value < 0.00001 ****), number of events (Figure 5C, control 35 ± 0.17/19.5 min vs. heptanol 4.77 ± 0.11/19.5 min, *p*-value < 0.00001 ****), amplitude of events (Figure 5D, control 2.34 ± 0.01 ΔF/F0 vs. heptanol 1.57 ± 0.02 ΔF/F0 *p*-value < 0.00001 ****), the duration of events (Figure 5E, control 13.32 ± 0.08 s vs. 13.09 ± 0.37 s, *p*-value < 0.0001 ***) and rise time (Figure 5F, control 4.03 ± 0.02 s vs. heptanol 4.95 ± 0.12 s, *p*-value < 0.0001 ***).

Heptanol severely affected the spread of calcium waves, since treated fish only showed non-spreading activity (1467 events 2.89 ± 1.75 µm), in contrast to untreated fish, which showed all three kinds of activities: non-spreading events (14188 events 0.43 ± 0.02 µm), short-distance events (127 events 32.21 ± 10.46 µm) and 33 long-distance events (138.71 ± 18.75 µm) Figure 5G.

Synchronic calcium events were eliminated by heptanol treatment as shown in Figure 5B,H, where black arrows indicate synchronic events in control condition; however, no events were detected under heptanol treatment. This plot also shows evidence of the reduction of the number of calcium events induced by heptanol, where 75 events per minute were detected in 244 ± 7.6 cells, in contrast to the control, in which 735 events per minute were detected in 2391 ± 39.24 cells. Finally, the maximal speed of calcium events was reduced from 40–60 µm/s in control fish to 15–25 µm/s in heptanol treated fish (Figure 5I). These data indicate that, at 5 dpf, gap junctions are fundamental for propagation of calcium waves and, upon heptanol-induced blockage, most of the activity detected comes from intracellular calcium stores.

## 3. Discussion

By employing a combination of morphometric analysis and live-cell calcium activity in zebrafish, we have identified critical events in the development of cerebellar radial glia. Specifically, we show that the morphology complexity of radial glia is related to the appearance of synchronic calcium events and calcium waves. We found that, at 3 dpf, most of the radial glia cells had two short processes and a small soma; in addition, calcium events were scarce, brief and did not propagate across cells. At 5 dpf, somas and processes were larger and calcium events began to appear and were more frequent and lasted longer. At 7 dpf, the cell processes retracted; however, there were more processes per radial glial cells while the calcium events remained constant.

Radial glial cells have been largely studied in the context of the origin of the complex primate brain cortex, where several morphological subtypes have been identified [34,35,36,37]. Our study shows that radial glia of the developing zebrafish cerebellum radically changes between 3 and 7 dpf. At 3 dpf, these cells are homogeneously distributed with their somas aligned along the midline and their processes project ventrally. Then, with the emergence of the CCe and the auricle at 5 dpf, new radial glial cells occupy these areas and reorganize due to an increase in the size of the cerebellum. At 7 dpf, the radial glia somas are aligned along the midline, but their processes project laterally. In this regard, it has been shown that spatiotemporal gene expression trajectories during maturation of human telencephalon gives rise to diverse radial glial cell types related to temporally and spatially restricted trajectories [38]. It will be interesting to determine how the transcriptomic landscape of zebrafish radial glia cells evolve during early development and how it compares to the juvenile and adult transcriptomic profiles [19,39].

The emergence of the morphological complexity of radial glia cell processes is temporally related to the origin of calcium activity. Gap junction coupling through somas and processes of neighboring cells is functional when the cerebellar cells layers are formed at 5 dpf. At this age, the somas and processes lengths almost doubled from 3 dpf, although the number of processes was transiently reduced (Figure 2). We found that the structures that form the cerebellum (field 1 and field 2) showed differences in their calcium activity, which might be related to different neuronal activity.

The expression of the basic molecular components that give rise to calcium communication in astrocytes and radial glia is already present at 3 dpf [40,41,42]; however, the activity is mostly local. Interestingly, the duration of calcium events at 5 dpf in the cerebellum (11.38 ± 0.43 s) is similar to that of astrocytes in the hindbrain at 6 dpf (12 ± 5 s) [43]. Calcium transmission is relevant for the generation and migration of new neurons during development [44,45], and here we have generated a model to study the involvement of calcium waves during early development, with the potential for applications in the study of neuropathogenic models.

## 4. Materials and Methods

### 4.1. Zebrafish Housing Care

Adults zebrafish (TABWIK) were used at 28 °C and under 14:10 light-dark cycles. The fish were bred and housed as reported before [46,47]. Protocols were approved by the Institutional Ethics Committee of the Institute of Neurobiology, UNAM (Protocol number. 95A approved on April 2016). 

### 4.2. Molecular Biology

Plasmid pTol2-gfap-GCaMP6s (Appendix A) was generated by substituting the elavl3 promoter of pTol2-elavl3-GCaMP6s (Addgene plasmid #59531 Watertown, MA, USA) for the gfap promoter form p5E-gfap (Addgene plasmid #75024, Watertown, MA, USA) by direct cloning using the Xho I (Thermofisher Scientific, Waltham, MA, USA) and Sal I (Thermofisher Scientific, Waltham, MA, USA) restriction sites. 

### 4.3. Generation of the Tg (pTol2-gfap-GCaMP6s)

Adult zebrafish were mated and the embryos collected in E3 medium (Hank’s medium with-HEPEs pH 7.4). Embryos were injected at 1-cell stage with 0.5–1.0 mixture 25 pg mRNA transposase and 25 pg DNA of plasmid (pTol2-gfap-GCaMP6s), 0.01% phenol red and then dissolved the mixture in 10 mM KCl (J.T. Baker, Phillipsburg, NJ, USA) [48,49].

After 24 h post-injection, embryos were screened under an epifluorescence microscope. Positive embryos were bred and, after 3–4 months, adults were mated to obtain the F1 offspring. F1 were screened by epifluorescence and genotyped by PCR (Appendix A). (Primers forward CTCGAGAACGTCTATATCAAGGCCG and reverse TCACTTCGCTGTCATCATTTGTACA).

### 4.4. Whole Mount Immunohistochemistry

We used a modified protocol as reported in [50]. Briefly, 1% tricaine (Sigma-Aldrich, Dorset, UK) were added to larvae at 3, 5 and 7 dpf for 10 min. Then, the larvae were fixed with 4% paraformaldehyde (Sigma-Aldrich, Dorset, UK) overnight at 4 °C, transferred to a glass container, washed for 5 min with phosphates-buffered saline (PBS) pH 7.4, then washed with ddH_2_O and permeabilized for 5 min in ice-cold acetone (J.T. Baker, Phillipsburg, NJ, USA). Blocking solution (PBS with 2% goat serum (Sigma-Aldrich, Dorset, UK) was added and incubated for 2 h at room temperature and washed 5 times for 10 min with PBS/BSA1% (Sigma-Aldrich, Dorset, UK) at room temperature. Primary antibody anti-GFAP (1:250, #GTX128741 GeneTex, Irvine, CA, USA) was added and incubated for 3 days at 4 °C. Then, larvae were washed 7 times for 10 min with PBS at room temperature. Secondary antibody anti-rabbit Alexa fluor 594 (1:500, #AB150080 Abcam, Cambridge, UK) was added and incubated for 2 days at 4 °C. Larvae were washed 7 times for 10 min with PBS at room temperature. DAPI (1:5000, D1306 Thermo Fisher, Waltham, MA, USA) was added and incubated for 1 h and then washed 7 times for 10 min with PBS at room temperature. Larvae were incubated in 25, 50, 75 and 100% glycerol (J.T. Baker, Phillipsburg, NJ, USA) for 20 min and finally, stored at 4 °C.

### 4.5. Confocal Imaging

For confocal images, whole larvae were mounted in slices with 2% low melting agarose (Sigma-Aldrich, Dorset, UK) in dorsal position and observed under Confocal Zeiss 780 LSM microscope (Plan-Apochromat 1 mm Korr DIC 25X, 0.8 NA). The lasers used were Argon A488, DPSS 561-10 and Chameleon Ultra-Ti:Sapphire 690-1064. For whole cerebellum imaging, a z-stack of 50–65 µm was processed with an interval of 1.3 µm for 1024 × 1024 pixels. 

### 4.6. Morphological Analysis 

Confocal z-stack (50–65 µm depth) from the caudal encephalon were processed using ImageJ/FIJI software (2.35 version, Bethesda, MD, USA) [51]. The regions of interest of the cerebellum were cropped, and the autofluorescence emitted by the skin and outer layer was removed by applying the tool “erase” of FIJI in every z-plane. Later, the GFAP-labeled and DAPI signals were split and processed independently. The GFAP-labeled background was subtracted (rolling ball radius, 60–70 pixels) and the general contrast increased by 0.1% saturated pixels. The local contrast was enhanced (CLAHE function, block size: 150–200, slope: 2) and the unsharp mask filter was applied (radius 1.0 mask weight 6.0). In the case of the DAPI signal, only the unsharp mask filter was applied. Finally, to visualize the radial glia during development, z-stacks were divided into three zones (ventral, medial and dorsal) and each one projected in a single plane (max intensity mode). The 3D reconstructions were built with the z-stacks processed using the 3D viewer plugin of FIJI. 

To determine the characteristics of radial glia according to the GFAP-labeled signal, we used the segmented line tool to draw the processes through pre-processed z-stacks and the freehand tool to draw the soma. Following this, each element was converted into a region of interest (ROI Manager Tool of FIJI). Total area and number and length of processes were determined (with the Measure Tool of FIJI) and the results were organized by dpf and cerebellar region [52].

### 4.7. Statistical Analysis of Radial Glial Cell Morphology

Descriptive statistics (mean, SD, SEM) of cell morphological parameters from larvae at 3, 5, and 7 dpf were obtained from N = 5 larvae per age. The difference between groups was determined by the Kruskal–Wallis test, followed by a post hoc Dunn’s multiple comparisons test. Differences between cerebellar region per age, were determined by a one-way ANOVA test followed by Tukey´s post hoc test. Finally, differences between central and lateral regions at 3 dpf were assessed by the Mann–Whitney U test. In all cases, we used an alpha, 0.05; results are reported as mean ± SEM, while *p*-value significance is reported as 0.10 (ns), 0.01 *, 0.002 **, >0.001 ***. The different statistical tests were chosen based on the type of distribution, variance homogeneity and number of samples.

### 4.8. In Vivo Calcium Imaging

The transgenic line (Tg gfap-GCaMP6s) was used for calcium images. Adults were mated and the embryos collected in E3 medium (pH 7.4). Pigmentation was avoided by adding 75 µm phenyl urea (Sigma-Aldrich, Dorset, UK) at 6 h postfertilization. Embryos were paralyzed with 0.5 mM tubocurarine (Sigma-Aldrich, Dorset, UK) at 3 dpf or 2 mM tubocurarine at 5 and 7 dpf for 10–15 min [53,54]. For imaging, embryos were mounted in melted 2% low melting agarose (Sigma-Aldrich, Dorset, UK) at 35 °C in a concave slide covered with 100 µL of E3 with tubocurarine [55,56]. 

For in vivo calcium imaging, an Olympus BX51W microscope was used (water-immersion objective 20×, 0.8 NA); the LED source was X-Cite XLED1 Excelitas Technologies at 465 nm, at 6% of power. The fish were kept in the dark at 26 °C during image acquisition. The fish were adapted for 20 min before recordings started. The videos were recorded at 2.55 or 4 Hz in fields 1 and 2 (Micromanager program with high resolution camera pco.edge 4.2, Kelheim, Germany).

### 4.9. In Vivo Pharmacological Blockage of Gap Junctions

5 dpf larvae were mounted in low melting agarose (2%), then 500 µm Heptanol (Sigma-Aldrich, Dorset, UK) freshly prepared was added for 30 min before image recordings. The fish were kept in the dark at 26 °C during image acquisition and images acquired at 2.55 Hz, 19.5 min. After finishing the recordings, the fish were returned to a water tank and the heart beat and blood flow were monitored for least 1 h; those with no activity were excluded from analysis.

### 4.10. Analysis and Statistics of Calcium Imaging

ImageJ/FIJI software was used to subtract background (Rolling ball radius 12 pixels) and the Turboreg (rigid body) plugin was used for motion correction. The cerebellum fields were selected as regions of interest. The changes in fluorescence in somas and processes were detected by an automated astrocyte quantitative analysis (AQuA) software program (https://github.com/yu-lab-vt/AQuA (accessed on June 2021)) [43] and a Matlab (version R2020a, United States) script was used for analysis (Appendix A). A nonparametric Kruskal–Wallis test was applied to assess whether samples originated from the same distribution at α = 0.05, followed by a comparison of the medias and their SEM with a multicomparation test α = 0.05; *p*-values were 0.05 *, 0.01 ** and 0.0001 ***. Differences between control and heptanol blocking test were determined by the Mann–Whitney U test at α = 0.05; *p*-values were <0.0001 ***, <0.00001 ****. For each parameter, the mean and SEM were reported (Appendix A). N = 6 larvae per age for each field (field 1 and field 2, at 4 Hz for 4.16 min). N = 6 per age in field 1 at 2.55 Hz (19.5 min). N = 6 for heptanol blocking test in field 1 at 2.55 Hz (19.5 min).

## 5. Conclusions

In this study, we focused on the relationship between the calcium events and morphological complexity of radial glia in the zebrafish cerebellum during early development. Radial glia were likely to be involved in many physiological processes that include their interaction with neurons and vasculature that, in turn, are associated with a variety of neurological diseases. We show that the morphological complexity of radial glia during the first week of development is remarkable, since it goes from simple stratified columns at 3 dpf to complex morphologies and interactions at 7 dpf. This transformation is related to increase in calcium activity from local signals that recruit a limited number of cells at 3 dpf to widespread calcium events and the appearance of calcium waves at 5 dpf, whose activity is inhibited by blocking gap junctions.

## Figures and Tables

**Figure 1 ijms-22-13509-f001:**
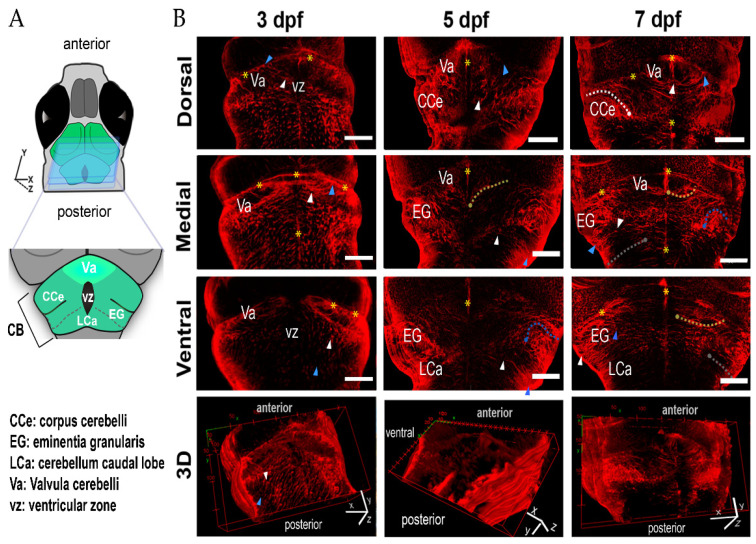
Radial glia cells during zebrafish development. (**A**) The scheme shows the regions of the cerebellum in the confocal images to the right. (**B**) Three representative optical sections of GFAP-labeled cerebella; radial glia are presented for each age: dorsal medial and ventral; the lower panels show the corresponding 3D reconstructions. In each plane, a representative soma (white arrowhead) and its end-feet (blue arrowhead) are depicted. Yellow asterisks indicate the midline and the continuum formed by the radial glia end-feet of the Va and CCe. Dashed lines show the trajectories of the radial glia processes. The position of the soma is indicated by a dot of the same color: Va, yellow dashed line; CCe, white dashed line; EG, blue dashed line; LCa, gray dashed line; dfp, days post fertilization; scale bars, 50 µm.

**Figure 2 ijms-22-13509-f002:**
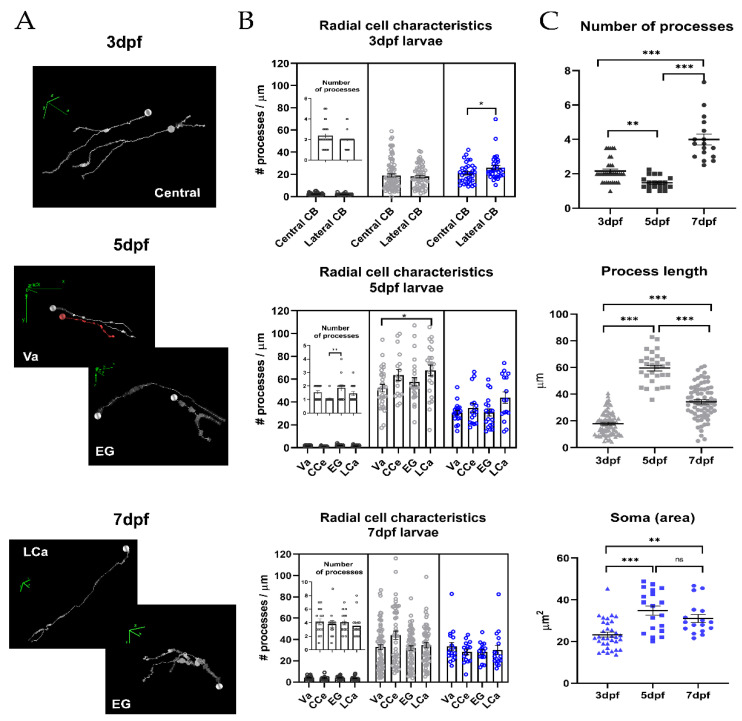
Morphometric characteristics of radial glia during cerebellar development. (**A**) Representative 3D reconstructions of radial glia randomly selected from different regions at 3, 5 and 7 dpf. The concentric rings along the process represent the spatial orientation and thickness of the process; in these representations, nuclei are indicated by a sphere, the processes in white and red lines and direction of the axis (x, y and z, green lines) (**B**) Nested graph showing the number (black dots), length (gray dots) of processes and soma area (blue dots, inset) in the different cerebellar regions at 3, 5 and 7 dpf (Va, CCe, EG and LCa; N = 5–7 larvae per stage). For simplification, only statistically significant differences are presented; cell somas at 3 dpf: central vs. lateral, 21 ± 1.5 µm^2^ vs. 26 ± 2.0 µm^2^, Mann–Whitney U test, (*p*-value 0.049 *, α 0.05), process length at 5 dpf: Va vs. LCa, 52.3 ± 3.3 µm vs. 67.7 ± 4.7 µm, ANOVA-Tukey (*p*-value 0.0309 *, α 0.05). (**C**) Graphs comparing the characteristics of radial glia from the whole cerebellum between the three ages evaluated (Kruskal–Wallis Test, Dunn’s post hoc, α 0.05, *p*-value: 0.1 (ns), 0.05 *, 0.01 **, <0.001 ***), graphs show mean ± SEM (see Table 2 for descriptive statistics).

**Figure 3 ijms-22-13509-f003:**
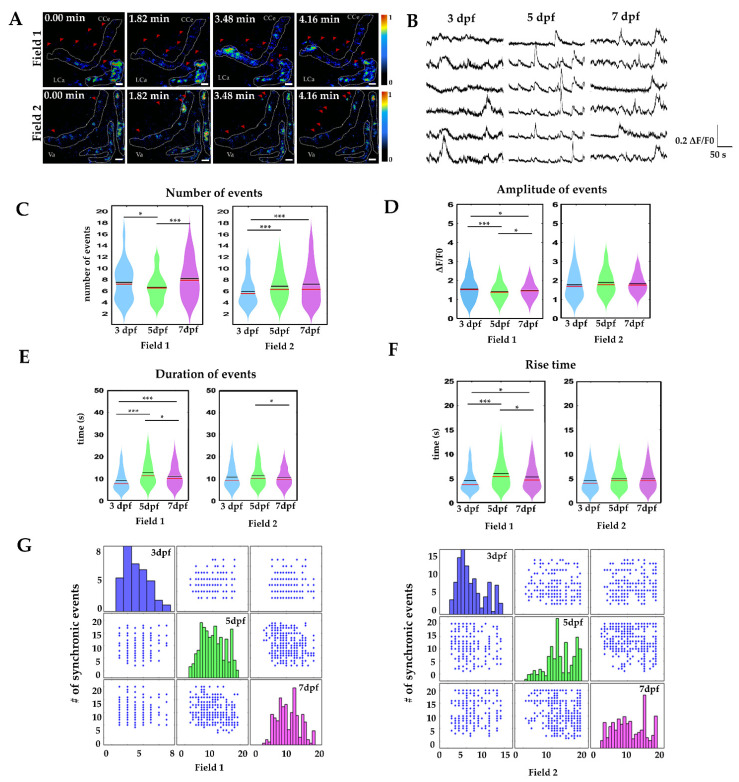
Calcium signalling during cerebellar development. (**A**) Calcium imaging in the lateral cerebellum at 5 dpf. A white dashed line delimits the radial glia somas; red arrowheads indicate activity in radial glia processes. (**B**) Sample recordings of calcium activity (ΔF/F0 during 4.16 min, at 3, 5 and 7 dpf in field 1). (**C**) Number of calcium events in 4.16 min. (**D**) Amplitude of calcium events (ΔF/F0). (**E**) Duration of calcium events (in seconds). (**F**) Rise time of calcium events (in seconds). The boxplots show data distribution, mean values (black line) and media (red line). Nonparametric statistics, Kruskal–Wallis test, α 0.05, *p*-value < 0.05 *, *p*-value < 0.001 **, *p*-value < 0.0001 ***. (**G**) Number of synchronic calcium events at 3 dpf, 5 dpf, 7 dpf (blue, green and magenta bars, respectively, scatter plot of x columns against y axes). For these experiments, at least 6 larvae per age and per field were recorded. Calcium events recorded per field 3 dpf *n* = 200, 5 dpf, *n* = 750, 7 dpf *n* = 400; scale bar 20 µm.

**Figure 4 ijms-22-13509-f004:**
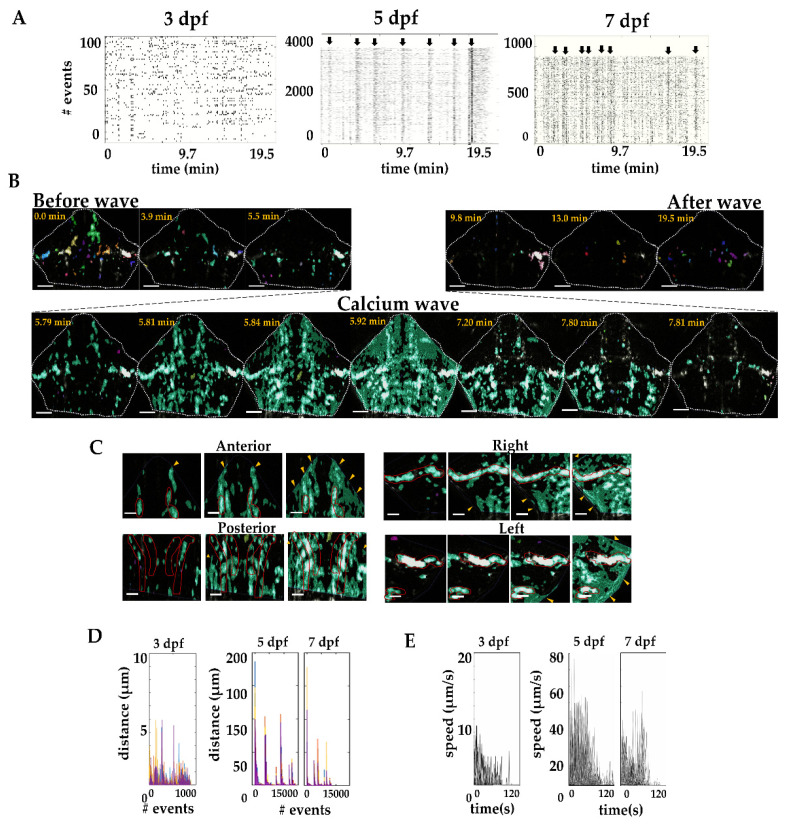
Calcium waves in the cerebellum. (**A**) Calcium events at 3, 5, and 7 dpf (each region of interest is aligned in time and each event is represented by a black dot, black arrows indicate the waves). (**B**) A calcium wave at 5 dpf. The calcium waves spread from the most posterior region of the cerebellum to the anterior region (white dashed line delineates the cerebellum). (**C**) Sample images of calcium activity spreading within radial glia in the lateral regions (left-right) and anterior and posterior regions. Red lines indicate the mid area where the radial glia somas are located, and yellow arrowheads indicate the pial surface where the terminal end-feet of the radial glia are located. (**D**) Spreading distance of calcium waves (as shown in (**A**)). At 5 and 7 dpf, only about 2% of the events propagated to form a calcium wave. (**E**) Calcium wave speed at 3, 5 and 7 dpf. 3 dpf, *n* = 897; 5 dpf, *n* = 14348; 7 dpf, *n* = 8848. N = 6 larvae per age. Scale bar 50 and 20 µm.

**Figure 5 ijms-22-13509-f005:**
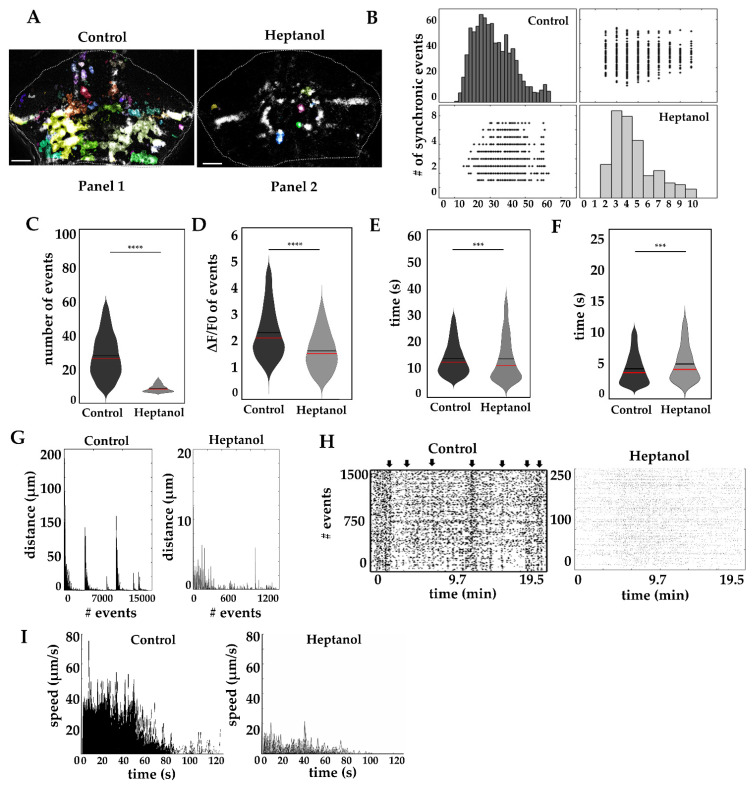
Calcium waves disruption in radial glia. (**A**) Calcium events in fish cerebellum after heptanol treatment (events with changes in fluorescence are shown with a multicolor mask, and the white dashed line delineates the cerebellum). Panel 1, long-distance spreading in control cerebellum, Panel 2 non-spreading events in heptanol treated fish. (**B**) Synchronic calcium events are reduced by heptanol treatment (black and gray bars, control and heptanol respectively. Scatter plot of x columns against y axes) (**C**) Number of calcium events in 19.5 min. (**D**) Amplitude (ΔF/F0). (**E**) Duration (in seconds). (**F**) Rise time (in seconds). (**G**) Propagation distance (in µm) of calcium events. (**H**) Raster plots of calcium waves (each region of interest is aligned in time and each event is represented by a black dot, black arrows indicate the time of occurrence of a calcium wave). (**I**) Maximal speed of calcium waves (µm/s). The box plots show data mean values (black line), media (red line) and data distribution. Non-parametric statistics, Mann–Whitney U test, α 0.05, *p*-value < 0.0001 ***, 0.00001 ****. Al recordings were obtained from 5 dpf larvae. 6 larvae were recorded for each condition. Calcium events recorded in control fish *n* = 14,348 and in heptanol *n* = 1467. Scale bar in A 50 µm.

**Table 1 ijms-22-13509-t001:** Characteristics of radial glia cells during cerebellar development.

Age	Number of Processes	Process Length (µm)	Area of Soma (µm^2^)
3 dpf	2.16 ± 0.12	17.89 ± 0.97	23.28 ± 1.19
5 dpf	1.48 ± 0.09	59.53 ± 2.17	34.87 ± 2.27
7 dpf	3.99 ± 0.31	34.37 ± 1.56	31.13 ± 1.90

**Table 2 ijms-22-13509-t002:** Characteristics of radial glial cells in the cerebellum.

Age	Number of Processes		Process Length (µm)		Area of Soma (µm^2^)	
3 dpf	Lateral ^1^	Central ^1^	Lateral ^1^	Central ^1^	Lateral ^1^	Central ^1^
	2.35 ±0.20	1.97 ± 0.12	19.10 ± 1.50	18.10 ± 1.24	21.00 ± 1.51	26.20 ± 2.03
5 dpf	Va ^1^	CCe ^1^	Va ^1^	CCe ^1^	Va ^1^	CCe ^1^
	52.2 ± 0.12	63.6 ± 0.06	30.7 ± 3.3	34.8 ± 4.9	21.2 ± 3.0	22.7 ± 4.1
	EG ^1^	LCa ^1^	EG ^1^	LCa ^1^	EG ^1^	LCa ^1^
	57.70 ± 0.23	67.7 ± 0.16	31.0 ± 3.7	43.9 ± 4.7	21.7 ± 3.4	28.9 ± 5.2
7 dpf	Va ^1^	CCe ^1^	Va ^1^	CCe ^1^	Va ^1^	CCe ^1^
	4.12 ± 0.43	3.77 ± 0.53	32.9 ± 2.4	44.1 ± 3.9	33.5 ± 3.8	28.4 ± 2.6
	EG ^1^	LCa ^1^	EG ^1^	LCa ^1^	EG ^1^	LCa ^1^
	4.06 ± 0.37	3.50 ± 0.44	32.1 ± 2.1	34.7 ± 2.5	28.4 ± 2.2	30.3 ± 4.4

^1^ Cerebellar region analyzed.

## Data Availability

https://github.com/epereida/dataset.

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
