# Peer review of "Calcium Signaling in the Cerebellar Radial Glia and Its Association with Morphological Changes during Zebrafish Development"

_ijms, 2021, doi:10.3390/ijms222413509_

Round 1

Reviewer 1 Report

See the

Review report

The authors in this study explored the relationship between the morphological changes and calcium signaling in the cerebellum of radial glia during early zebrafish development. The larvae were divided into three groups during development: 3, 5 and 7 days post-fertilization. They found that radial glia were bigger in terms of soma size, process length and numbers 7 days after fertilization, compared to 3 and 5 days. The spontaneous calcium events appeared on day 5 when radial glia showed more complex changes in morphology. These calcium elevations happened before the formation of neuronal circuit. These findings might provide insights into the development of some neurological diseases. However, there are some concerns below.   

Major concerns:

1 Line 169, I am concerned about the data here in this paragraph. For example, 7.4±4.6/4 min vs 6.12±4.1/4 min, p-value<0.001. 4.6 and 4.1 seem too big as SEM for their mean 7.4 and 6.12, respectively.

2 Figure 2A, any reason the authors chose 3.48 min for field 1 but 1.82 min for field 2?

Minor concerns:

1 Line 68, when the neuronal circuit would be formed? Or any reference?

2 Line 73, have the authors tried day 10 post-fertilization? Since their results showed different morphometric changes between day 3 to day 5 and day 5 to day 7.

3 figure 1, maybe just show one representative soma and its end-feet? Too many white arrowheads are confusing.

4 Table 1, is it necessary to show the std deviation when the authors already had the SEM?

5 Please use full names (CB and RG) in the main texts as much as you can. This will make it easier for readers.

6 English editing is highly recommended. Some grammar mistakes:

a line 31, suggest…

b line 14

attachment. 

Reviewer 2 Report

In this work, Pereida-Jaramillo and collegues used a combination of whole mount immunofluorescence and calcium imaging in gfap-GCaMP6s-transgenic zebrafish determining the contribution of calcium waves in the morphological changes involved in the cerebellum development.

The manuscript is very exhaustive in term of content, methods description and collection of data. For this reason, I fully support the publication of this manuscript in this journal.

Reviewer 3 Report

The authors studied the calcium signaling in the cerebellar radial glia of zebrafish and reported the association with morphological changes during development. The study is well performed, the results are interesting based on evidence, the data was presented clearly, and the manuscript was well organized. 

One suggestion on the "4.7. Statistical analysis" part:

There are several different tests were used for data analysis. It will be more convincing if the authors clarify what is the rationale for choosing each different test.

Reviewer 4 Report

Review of manuscript by Pereida-Jaramillo et al. entitled “Calcium signaling in the cerebellar radial glia and its association with morphological changes during zebrafish development”. The work is very interesting and the data from the transgenic line Tg(gfap::GCaMP6s) is very important. However, study lack important experiments to prove the concept in detail.

There are following concerns:

  1. What is the source of calcium? Is it intracellular stores or influx from extracellular space? This need some in-vitro and/or ex-vivo experiments.
  2. What is the mechanism for calcium wave propagation? Need some blocking experiments to confirm.
  3. How do you confirm the relation between higher calcium levels and increased RG morphological complexity? This needs calcium blocking study, to show calcium blocking reverse/inhibit RG complexity.

Round 2

Reviewer 1 Report

The authors have addressed all my concerns.  

Reviewer 4 Report

Authors have satisfactorily revised the manuscript. I highly appreciate their effort that they took time to incorporate some more experiments which eventually enhanced the quality and clarity of the manuscript. No further revision is required from my end. Some minor editing or spell check required.